# Effects of Two Commercial Diets on Several Reproductive Parameters in Bitches: Note One—From Estrous Cycle to Parturition

**DOI:** 10.3390/ani11010023

**Published:** 2020-12-25

**Authors:** Riccardo Orlandi, Emanuela Vallesi, Serena Calabrò, Alessandro Vastolo, Nadia Musco, Alessandro Troisi, Angela Polisca, Pietro Lombardi, Monica I. Cutrignelli

**Affiliations:** 1Tyrus Veterinary Clinic, Via A. Bartocci 1/G, 05100 Terni, Italy; riccardo.orlandi83@hotmail.it (R.O.); manu0391@libero.it (E.V.); 2Department of Veterinary Medicine and Animal Production, University of Napoli Federico II, 80100 Napoli, Italy; serena.calabro@unina.it (S.C.); alessandro.vastolo@unina.it (A.V.); pietro.lombardi@unina.it (P.L.); monica.cutrignelli@unina.it (M.I.C.); 3School of Bioscience and Veterinary Medicine, University of Camerino, 62024 Matelica, Italy; alessandro.troisi@unicam.it; 4Department of Veterinary Medicine, University of Perugia, 06124 Perugia, Italy; angela.polisca@unipg.it

**Keywords:** dog, estrous cycle, pregnancy, vaginal cytology, follicular development, fetal resorption

## Abstract

**Simple Summary:**

Reproductive efficiency is a key aspect of all breeding species. Several factors, such as infective diseases, hormonal, and nutritional status could affect female fertility. As demonstrated by high numbers of newborns per litter and proper fetal development, nutritional status at mating, and correct nutritional management during pregnancy are fundamental. In this study, two diets with different ingredients (protein and lipid sources) and different macro and micronutrients concentrations (crude protein, fatty acids profile, vitamins, and mineral concentration) were administered from two months before the expected onset of proestrus to parturition to 18 bitches divided into two groups. The experimental diet, richer in protein, essential and polyunsaturated fatty acids and vitamins, when compared to control diet, seems able to guarantee better clinical presentation of estrus and embryonic development in the first third of pregnancy with a lower incidence of a fetal resorption.

**Abstract:**

The close link between nutrition management and reproductive efficiency is well known, but there is very little data available concerning this topic in canine species. The present study aimed to compare the effect of two different diets upon the follicular period and gestation in bitches. Eighteen pluriparus medium and large size bitches were recruited and divided into control (CTR) and experimental (EX) groups and fed, respectively, with a commercial kibble diet and a specially formulated diet from two months before the expected onset of proestrus up to the end of the trial. It was possible to observe how the EX group had a better clinical presentation of the estrous phase, a higher number of ovarian follicles (*p* < 0.05), a lower percentage of fetal resorption (*p* < 0.05), and lower oxidative status, expressed by d-ROMs (*p* < 0.01), at the moment of pregnancy diagnosis compared to CTR group. Moreover, the EX group showed a lower fetal resorption rate and higher litter size (*p* < 0.05). These results highlight how a diet characterized by high protein and fat content and richer in essential fatty acids can improve reproductive performance in dogs.

## 1. Introduction

Nutrition plays a crucial role in satisfying metabolic demands during pregnancy. The maternal diet must provide sufficient energy and amounts of nutrients to satisfy the mother’s requirements, as well as the needs of the growing fetus, and enable the mother to lay down stores of nutrients for fetal development and for lactation [1]. Proper nutrition management for reproducing bitches should precede mating and continue throughout gestation and lactation in order to improve maternal and fetal requirements and health. Within the first two-thirds of the period of gestation, energy requirements should not increase [2], while they rise to 1.25–1.50 times maintenance after the 35th–40th days [3]. From this last period up to parturition, due to the volume occupied by the uterus inside the abdomen and to the hormonal status, voluntary feed intake decreases, and several small meals are recommended to be administered daily [2,3].

The use of a commercial diet, specially formulated for canine pregnancy, is linked to the necessity to guarantee an efficient reproductive capacity in the bitch and optimal growth rates in puppies [2]. Essential fatty acids, vitamins, and minerals (particularly calcium, phosphorus, and magnesium) play an important role in ovarian hormone and uterine protein production, placentation, and fetal development and, when correctly balanced with other micro and macronutrients, may improve early embryonic survival, litter size, enhance lactation and neonatal survival [4,5].

Conversely, inadequate nutrition management, characterized by either deficiencies or excesses, may have a detrimental effect in pregnancy even on healthy animals. Indeed, both conditions have been shown to exert negative effects on fertilization rate and number of fetuses [4,6]. Moreover, a poor body condition score could also induce metabolic disturbances such as gestational ketosis, impaired endocrine equilibrium, impaired placentation, increased neonatal death rates, and eclampsia, while obesity is known to contribute to the development of dystocia [6,7].

Despite the central role of nutrition in improving reproductive efficiency and avoiding maternal and fetal pathologies during pregnancy, very little data regarding canine species are available and, consequently, many aspects related to the connection between reproduction and nutrition are still little understood [8,9].

The aim of the present study was to compare two complete diets for bitches different for ingredients and nutrients content. Each diet was administered to bitches of medium and large size from two months before the expected onset of proestrus up to parturition. The effects on estrus vaginal cytology, follicular development, and pregnancy progression are reported in this paper.

## 2. Materials and Methods

All the procedures used in the study were approved (PG/2020/0044625) by the Ethical Animal Care and Use Committee of the University of Naples Federico II in accordance with local and national law regulations and guidelines (Dlgs 26, 4 March 2014).

### 2.1. Animals and Diets

For this study, 22 privately owned pluriparus bitches (mean: 1.5 previous pregnancies) at two months before the expected onset of proestrus were recruited from a private veterinary clinic. The canine breeds of the dogs were represented: Bernese Mountain dog (2), French Bulldog (2), English Bulldog (3), Dobermann (1), Argentine Dogo (2), Pitbull (2), Great Danes (2), Golden Retriever (1), Hovawart (2), Labrador Retriever (2), German Shepherd (1) and Whippet (2). All dogs were judged healthy based on routine clinical examination and hematological exams. According to the clinical history of each subject, the inter-estrus intervals of previous cycles were judged normal. All subjects were of proven fertility with a history of previous normal pregnancies. Moreover, the studied population was negative for Brucellosis and was vaccinated against Herpesvirus.

The dogs were equally divided into two groups (11 bitches) and each group was fed one of two diets which differed as regards their ingredients (starch and protein sources), energy, protein and folate levels. The two groups of bitches were given the names Control_CTR and Experimental_EX (Table 1).

The ingredients of the two diets were:CTR: poultry meal, rice, corn, poultry fat, fishmeal, beet pulp, fish oil, seeds oil, sodium chloride and dehydrated Saccharomyces cerevisiae.EX: poultry meal, spelt, oat, potato, fishmeal, hydrolyzed fish protein, beet pulp dehydrated eggs, fish oil, flax seed, calcium carbonate, monocalcium phosphate, potassium chloride, psyllium, MOS, sodium chloride and dehydrated Saccharomyces cerevisiae.

Both diets were supplemented with a mix of vitamins and microminerals in proportion to their specific energy levels. In addition, both diets were supplemented with different levels of folate (CTR 12.84 and EX 13.14 μg/MJ Metabolizable Energy ME).

During the trial, four owners decided to drop out of the protocol, and this study, so we report only the data of the bitches that concluded the protocol. Consequently, the experimental groups were distributed as follows:Control group (CTR): composed of 4 bitches of medium size and 5 of large size (body weight 31.94 kg ± 13.08; BCS, 5 points scale: 3.30 ± 0.26);Experimental group (EX): composed of 2 bitches of medium size and 7 of large size (body weight 29.08 kg ± 14.35; BCS, 5 points scale: 3.63 ± 0.52).

Both the diets were produced by Farmina Pet-foods (Nola, Italy). The first was a commercial kibble diet formulated for adult dogs, whereas the second was specifically formulated in order to perform the trial.

Each bitch group was fed one of the diets during all the trial, and the daily individual rations were calculated according to NRC [10] in function of body weight, stage of pregnancy and body condition score. During the first two-thirds of pregnancy, the daily ratio was divided into 2–3 meals, and in the last third of gestation, the ratio was left at the bitches’ disposal all day long.

### 2.2. Monitoring of Bitches until Pregnancy Diagnosis

The owners were instructed to measure some parameters of the bitches at home, such as cardiac and respiratory rate and rectal temperature once a week starting from two months before the expected onset of proestrus until the weaning of the puppies.

From the first appearance of vulvar serosanguineous discharges, indicating the onset of proestrus, the cycle was monitored at veterinary clinic by means of colpocitology, serum progesterone and ultrasound of the ovaries every 3 days during proestrus and estrus in order to detect ovulation. After 48 h from ovulation, the bitches were mated twice 24 h apart. Ultrasound examination was carried out every 3 days from the onset of proestrus until the day of ovulation in order to evaluate the follicular development and, then, to confirm pregnancy on day 18 post-ovulation and, from this time, repeated twice a week to evaluate embryonic development and viability until parturition.

### 2.3. Vaginal Cytology and Plasma Progesterone Determinations

From the onset of proestrus, vaginal smears, obtained from the cranial portion of the vagina, were taken every 3 days until the day of ovulation from each bitch and promptly examined using hematoxylin-eosin staining as described by Wright [11]. The evaluation of the vaginal smears was performed using a light microscope (Nikon eclipse E200, Nikon Instrument Europe Amstelveen, Amsterdam, The Netherlands) at a magnification of 100× to 400× as described Antonov [12]. At the same time, serum progesterone concentrations were determined by enzyme linked fluorescent assay (Minividas Biomerieux, Marcyl’Etoile, France). According to Concannon [13], the ovulation time was identified by elevated serum progesterone concentrations (i.e., >5 ng/mL) and a preponderance of superficial cells in the vaginal cytology.

### 2.4. Ultrasound Scanning

All ultrasonographic examinations were performed on un-sedated bitches positioned in right lateral recumbency using a GE Logiq E9 (GE Medical Systems, Milwaukee, WI, USA) ultrasound machine equipped with 9 L linear transducer (5–9 MHz) coupled via acoustic gel which was applied directly to the shaved abdominal wall.

From the onset of proestrus, the ultrasound was performed every 3 days until the day of ovulation in order to evaluate the ovaries. More particularly, the gonads were detected behind the caudal pole of the ipsilateral kidney and the number and size of the follicles present were recorded.

The time of ovulation was determined when, compared to the previous ultrasound scanning, a clear transformation of the image of the ovaries was recorded and at least some of the follicles present previously had collapsed, as described by Fontbonne [14].

Eighteen days after ovulation, ultrasonographic scans were performed and repeated twice a week throughout the pregnancy until parturition. At each examination, the following parameters were considered in order to evaluate embryo/fetal viability and development: inner chorionic cavity diameter, crown-rump length, body diameter, biparietal diameter, fetal heart rate and the appearance of an embryo/fetal organs to the gestational phase (Figure 1).

### 2.5. Weight Control and Oxidative Status

The bitches’ body weights and body condition scores (BCS, scale 5 points) were recorded at the time of recruitment and then weekly. At the moment of recruitment, the bitches were subjected to blood sampling in order to perform biochemical analysis by means of an automatic biochemical analyzer AMS AUTOLAB (Rome, Italy) using reagents from Spinreact (Santa Colomna, Spain) to determine: blood urea nitrogen (BUN), creatinine (CREA), glucose (GLU) total proteins (TP), albumin (ALB), bilirubin (BIL), aspartate amino transferase (AST), alanineaminotransferase (ALT), gamma-glutamyltransferase (GGT), cholesterol (CHO) and triglycerides (TRI), chloride (Cl) sodium (Na), calcium (Ca) and phosphorus (P) (data reported as Appendix A).

Moreover, blood samples were collected at the moment of pregnancy diagnosis in order to evaluate the reactive oxygen metabolites (d-ROMs test) and biological antioxidant potential (BAP test) on serum aliquots which were measured using reagents from Diacron International s.r.l. (Grosseto, Italy) validated for canine species.

### 2.6. Statistical Analysis

The data was statically analyzed by means of one-way ANOVA according to the following model:y_ijk_ = μ +T_i_ +S_j_ + (T × S)_ij_ + ε_ijk_
where y is the dependent variable, μ is the mean, T is the treatment effect (i: CTR, EX), S is the dog size effect, j: medium (body weight < 30 kg) and large (body weight > 30 kg), T × S is the first level of interaction and ε is the error effect. When significant differences were found in the ANOVA, means were compared using Tukey’s test.

Fetal resorption and litter size data were analyzed using the Wilcoxon non-parametric test.

All the statistical analyses were performed using JMP 14 software, SAS Institute, Cary, NC, USA.

## 3. Results

### Monitoring of Bitches until Pregnancy Diagnosis

Nutritional treatment affected only a few parameters of ovarian ultrasound and vaginal cytology (Table 2 and Table 3). During anestrus and ovulation, the long ovarian axis was significantly (*p* < 0.05) higher in the EX group compared to CTR one. The same trend (*p* < 0.05) was observed for follicles per ovary during estrus. During estrus, the experimental group showed significantly (*p* < 0.05) higher values of progesterone and a higher percentage of superficial cells.

The size of the bitches significantly (*p* < 0.05) affected only a few parameters. Particularly, bitches of medium size showed higher values of short ovarian axis and progesterone level during proestrus. During estrus, bitches of large size showed a significantly (*p* < 0.05) higher percentage of superficial cells than medium size.

Figure 2, Figure 3, Figure 4 and Figure 5 showed the rate of embryonal growth (e.g., vesicle, biparietal and body diameters and crown-rump length). Each parameter showed a different tendency in function of nutritional treatment and the bitches’ size:Inner chorionic cavity diameter—EX group showed higher values than CTR one and, in particular, on the 23rd day the difference was significant (1.30 vs. 0.80 cm; *p* < 0.05). Regarding size, the larger bitches showed higher values with a significant difference on the 23rd day (1.27 vs. 0.83 cm; *p* < 0.05) compared to bitches of medium size.Biparietal diameter—both effects (diet and size) did not affect this parameter, except for the 37th day when the diameter value of CTR group was significantly higher (1.36 vs. 1.13 cm, *p* < 0.05) than EX group one.The body diameter of the control group showed higher values than the experimental group even if the difference was significant only on the 41st day (1.91 vs. 1.61 cm, *p* < 0.05). Concerning size, until the 46th day, large bitches showed a higher diameter compared to bitches of medium size, while from the 46th day the values became similar.Crown-rump length—the control group showed a greater length during the first 50 days of gestation (50th: 10.74 vs. 9.75 cm, *p* < 0.05), afterward the values became overlapped. The embryos of large size bitches were always longer than those of medium-sized bitches.

All dogs showed pregnancy duration (63 ± 2.0 days) that falls in average values [11,13] with the exception of the five brachycephalic subjects for whom a cesarean section was planned, all the parturitions were natural. The fetal resorption (Figure 6) recorded by ultrasound checks was significantly (*p* < 0.05) different for EX and CTR group (7.05% and 11.94%, respectively), and the number of puppies per litter (Figure 7) was significantly (*p* < 0.05) higher for EX group than CTR one (9.11 vs. 3.89, respectively). No malformations were observed in newborns.

Clinical data collected on respiratory rate, heart rate and rectal temperature was normal in all subjects for the entire study period. No differences between groups were found for bitches’ body weight (Figure 8) after parturition (32.56 vs. 39.73 kg for CTR and EX group, respectively) even if the BCS (Figure 9) at the end of pregnancy was significantly higher (*p* < 0.001) for EX group (3.42/5) than for CTR one (2.75/5).

The oxidative statuses of the bitches, which were registered at pregnancy diagnosis, were affected both by nutritional treatment and size (Figure 10): in particular, d-ROMs values were higher (*p* < 0.01) in CTR group than in the EX one and in medium-sized bitches as compared to large ones (*p* < 0.05). In any case, no significant differences were observed for BAP values, even if the trend was opposite to that observed for d-ROMs.

## 4. Discussion

Poor nutrition, either deficiencies or excesses, adversely affects pregnancy even in healthy animals. Excellent nutrition can improve early embryonic survival, improve litter size and birth weight, and neonatal survival [6,7]. Macro and micronutrients have various effects on ovarian hormone production, uterine protein production, placentation and fetal development. To take full advantage of the potential benefits of good nutrition, a diet appropriate for pregnancy and lactation should be fed from the time of breeding onward [15].

The Association of American Feed Control Officials [16] and the European Pet Food Industry Association (FEDIAF) [17] have established minimal requirements for the constituents of diets formulated for reproduction. This ensures adequacy, but not necessarily optimal nutrition. The quality and chemical form, as well as the quantity, of the constituents, are important. Therefore, the results of feeding trials are more important than the stated “diet formulation”.

Overall, both the experimental and control group of bitches enrolled in the follicular phase progressed normally. The weight and body condition scores registered during pregnancy for all bitches fall into the range indicated as optimal for the species, with a weight gain of 15–25% of the initial body weight before whelping [2,4,5].

From a clinical point of view, the greater number of superficial cells and the higher value of progesterone found in the EX group during estrous make this phase more typical and cytologically easy to recognize as is referred to in the literature [18]. Moreover, the higher number of preovulatory follicles found in the bitches fed with EX diet suggests a potential increase in fertility linked to the higher allowances of macro and micronutrients. The trend was confirmed by the higher number of puppies per litter registered in EX group compared to CTR one.

Resorption is defined as early embryonic or fetal death within the first 45 days of pregnancy [15]. When resorption occurs in a bitch, there is absence of external signs. There are no contractions, and the fetus is not being expelled out, but the fetus is dissolved into the uterus without externally visible signs. As reported by Sharma et al. [19] the size of the breed was found to have a significant effect on the incidence of fetal resorption in bitches, the incidence being significantly higher in medium and large sized breeds. However, in our study, no differences between the bitches’ size were detected for litter size. Gaytàn et al. [20] reported that the bitches found a significant effect (*p* < 0.01) of % vaginal cornified epithelium at the moment of artificial insemination and litter size: the bitches with >75% of vaginal cornified epithelium showed greater (*p* < 0.01) whelping rate (85.1% vs. 51.6%, respectively). Considering that in our study we found significant differences (*p* < 0.05) for the number of vaginal superficial cells comparing the groups (76.8% vs. 67.0% for EX and CTR group, respectively) and for litter size (9.11 and 3.89, for EX and CTR group, respectively), it is possible that the availability of the different nutrients could affect both parameters.

In any event, both the fetal resorption and number of puppies per litter registered in both groups were in the physiological ranges for the species, indicating an adequate intake of nutrients with both diets [21].

The progesterone profile through the follicular period was normal, and the difference within groups, found at the beginning of estrus, can be linked to the broad variability in the rise of this hormone [17]. In contrast to Hollinshead and Hanlon [22], we found a significant (*p* < 0.05) difference of progesterone levels in proestrus between medium and large-sized bitches.

The better reproductive outcome registered in EX group could be related to the higher supplementation of essential fatty acids of EX diet. Indeed, their deficiencies are responsible for host symptoms and disorders, including reproductive disorders, newborns abnormalities in kidney and liver; for this reason, essential fatty acids must be present in pregnant dog diet to ensure that hormones production and egg cells development. Indeed, increased essential fatty acids content led to an increase in Arachidonic acid in phospholipids of ovarian follicular granulosa cells stimulating prostaglandin production in ovarian follicular cells and prostaglandins, in turn, are known to stimulate ovarian steroidogenesis [23].

In dog Linoleic (LA, C18:2 n6) and α-Linolenic (ALA, C18:3 n3) are considered essential fatty acids (EFA) because they cannot be synthesized ex novo in the body end consequently the dogs have to assume LA and ALA with the diet. These EFA are considered the main precursors of n6 and n3 fatty acids, such as EPA and DHA. Moreover, as suggested by Bauer [24], there is a competition between LA and ALA metabolism. In particular, these authors compared the effect on coats and skin of two iso-energy and iso-fat diets formulated with different fat sources: sunflower seed (rich in omega-6 linoleic acid, LA) vs. whole ground flaxseed (rich in omega 3, ALA). They observed a significantly higher dermatological improvement using flaxseeds and postulated a sparing effect of ALA on LA. Considering the similar fat content of the diets administered to EX and CTR group in this study, it seems possible that the better performance obtained by EX group could be due to this effect. Indeed, CTR diet was characterized by a higher LA:ALA ratio (29.87 vs. 16.44 in CTR and EX groups, respectively). Moreover, the NRC [10] recommends an LA:ALA ratio of between 2.6 and 16.0.

In the first third of gestation, the diameter of the embryo vesicles was greater in the EX group, suggesting a faster growth rate at the beginning of gestation related to the experimental diet. These results could be due to the higher protein content of EX diet compared to CTR one (17.90 and 19.73 g/MJ ME) [24].

During pregnancy, different percentages of fetal resorption were recorded between groups (11.94% vs. 7.5% for CTR and EX group, respectively). The percentages of fetal death recorded in this trial are closer to the lower percentages commonly reported in the literature and ranged from 10 to 15% [25,26]. Our study population gives rise to some considerations. In fact, among the causes of embryo/fetal resorption, the following must be considered: infectious diseases, physical trauma, hormonal dysfunction, fetal abnormalities, or genetic and chromosomal alterations [14,25]. However, many of these causes can be excluded in our case considering the history and the method of recruitment used for the study. In this context, taking into account that during pregnancy there is a high demand for energy, protein and vitamins which may affect the survival of the embryos if not available [27], the low percentage of fetal resorption of the experimental group should be related to the specific diet, even if it is difficult to define which nutrients (e.g., crude protein, LA:ALA, folate) play a more decisive role. Indeed, nutritional factors were considered a non-infectious potential cause of fetal resorption [19]. Because dietary ingredients vary in protein content (range, 27–34% as fed) and amino acid profile, diets with 29–32% animal-based protein sources are recommended for pregnant bitches [28]. Similarly, both the overall fat content and the fatty acid profile affect pregnancy rates, litter size and the number of newborns. Conversely, obesity is known to contribute to the development of dystocia and increased neonatal mortality [15]. Despite the calcium and phosphorous content of the experimental diet was double compared to CTR diet, we did not observe significant differences in terms of serum calcium and phosphorous levels between groups. These are probably ascribable to the superimposable Ca/P ratio of the diets and/or to the high ability of the bitches to regulate the deposition and mobilization of calcium for milk production [29]. Considering that our observations were limited to blood profile analysis, we are not able to exclude long-term bone demineralization phenomena due to the dramatically elevated calcium requirements at the initiation of lactation.

The oxidative status of the bitches measured at the moment of pregnancy diagnosis testifies a good antioxidant action of reactive species and/or a decrease of the efficiency of antioxidant system [30]. Indeed, the d-ROMs test provides a measure of the whole oxidant capacity of plasma and its decrease is considered an indicator of cellular health. Even if the d-ROMs levels observed in both groups fall into the normal range of values for the canine species proposed by Sechi et al. [30] and Pasquini et al. [31], EX group showed significantly lower values compared to CTR ones, indicating a general decrease of reactive oxygen species (ROS) production. This result could be related to the dietary factors involved in the prevention of the negative effects of oxidative stress, such as feed manufacturing and storage conditions, use of antioxidant supplements [32]. Indeed, both diets used in this trial were supplemented with tocopherols as antioxidants; the ether extract concentration of the diets was similar although the fatty acid profile was different. The EX diet showed higher concentrations of Linoleic, α-Linolenic acids and Eicosapentaenoic acid (EPA) plus Docosahexaenoic acid (DHA). This could be due to the presence in EX diet of flaxseeds which were rich in Linoleic, α-Linolenic acids and in γ-tocopherol, which preserves cells from oxidation [33].

## 5. Conclusions

It is possible to highlight that the administration of the experimental diet allowed an improvement of the clinical presentation of estrus, an increase in the number of ovarian follicles, a faster rate of early embryonic growth, a lower percentage of fetal resorption and a higher litter size. Despite the non-homogeneous distribution between the groups of bitches of large and medium size, which represents the limit of this trial, our results confirm how a diet characterized by a different distribution of protein, fat and carbohydrates and a diet richer in essential fatty acids can improve dog’s reproductive performances.

Further studies are needed to better identify the interaction between the different components of the diet and a peculiar metabolic phase such as canine pregnancy.

## Figures and Tables

**Figure 1 animals-11-00023-f001:**
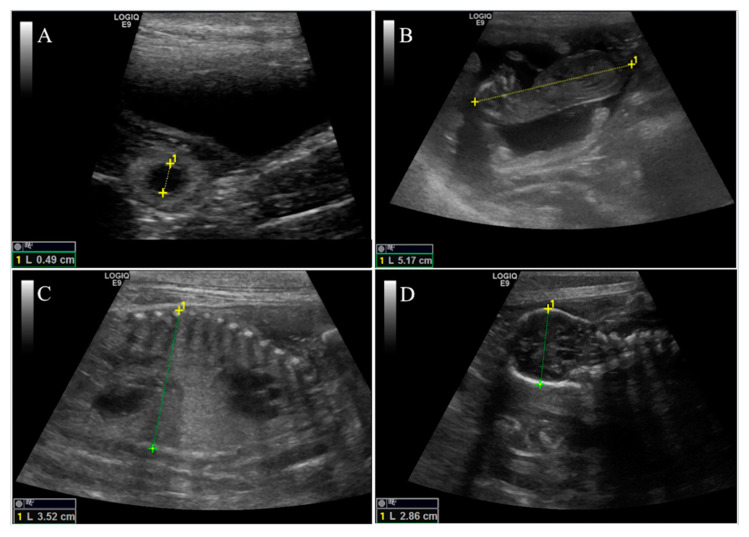
Representative ultrasonographic scans of biometry measurements recorded during pregnancy in an Argentine Dogo. (**A**): inner chorionic cavity diameter at 23 days of pregnancy. (**B**): crown-rump length (distance from the top of the head to the base of the tail) at 40 days of gestation. (**C**): body diameter in the transverse plane at the level of fetal liver at 53 days of gestation. (**D**): biparietal diameter (distance between parietal bones) at 57 days of pregnancy.

**Figure 2 animals-11-00023-f002:**
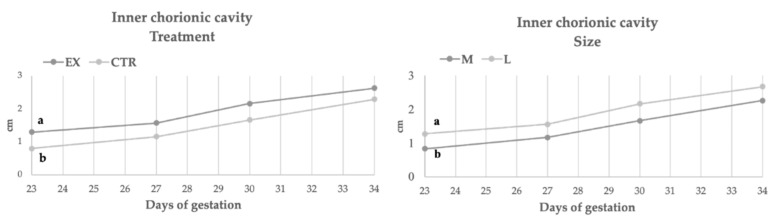
Inner chorionic cavity during gestation in function of diets and size. EX: experimental group CTR: control group; a, b: *p* < 0.05.

**Figure 3 animals-11-00023-f003:**
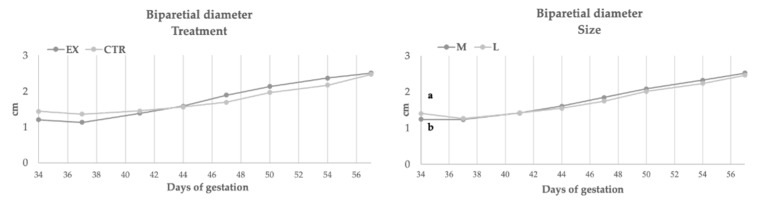
Biparietal diameter during gestation in function of diets and size. EX: experimental group CTR: control group; a, b: *p* < 0.05.

**Figure 4 animals-11-00023-f004:**
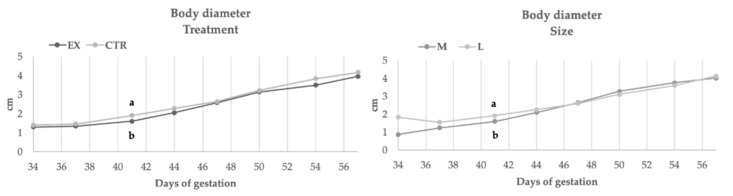
Body diameter during gestation in function of diets and size. EX: experimental group CTR: control group; a, b: *p* < 0.05.

**Figure 5 animals-11-00023-f005:**
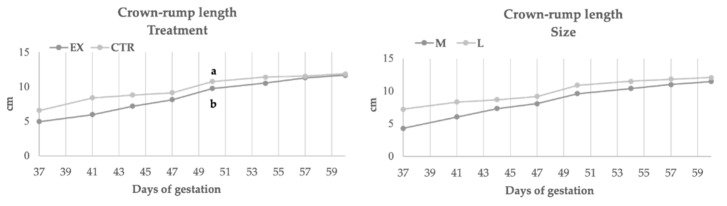
Crown-rump length during gestation in function of diets and size. EX: experimental group CTR: control group; a, b: *p* < 0.05.

**Figure 6 animals-11-00023-f006:**
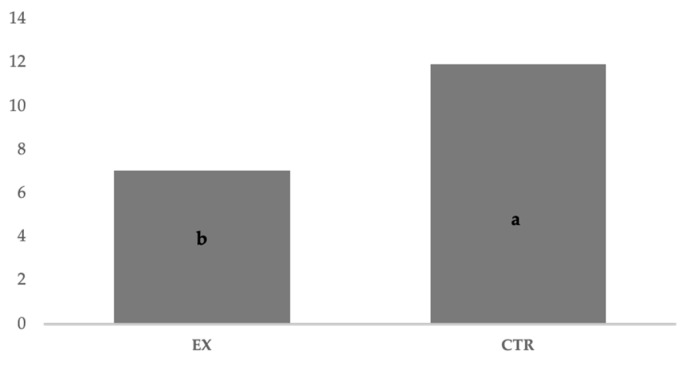
Percentage of fetal resorption recorded by ultrasound checks between two groups. EX: experimental; CTR: control; a, b: *p* < 0.05 (Wilcoxon non-parametric test).

**Figure 7 animals-11-00023-f007:**
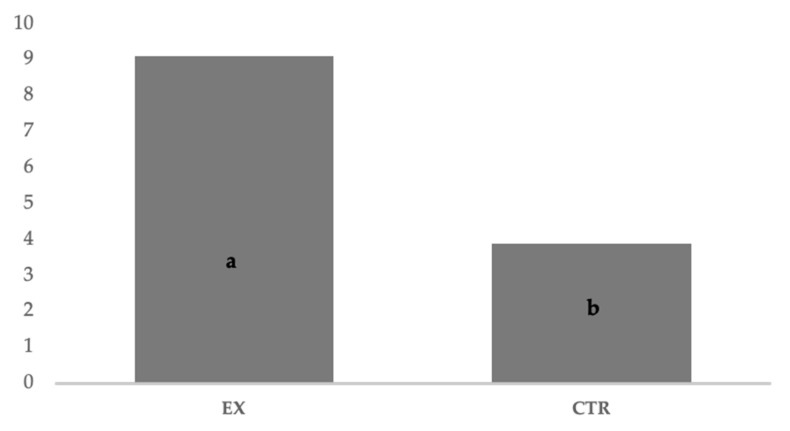
Number of puppies per litter recorded into the groups. EX: experimental; CTR: control; a, b: *p* < 0.05 (Wilcoxon non-parametric test).

**Figure 8 animals-11-00023-f008:**
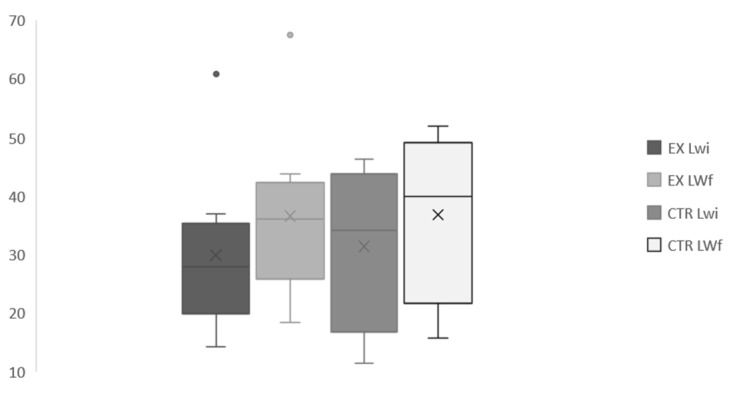
Bitches body live weight at the beginning and at the end of pregnancy. EX: experimental; CTR: control; LW_i_: initial body weight; LW_f_: body weight after delivery.

**Figure 9 animals-11-00023-f009:**
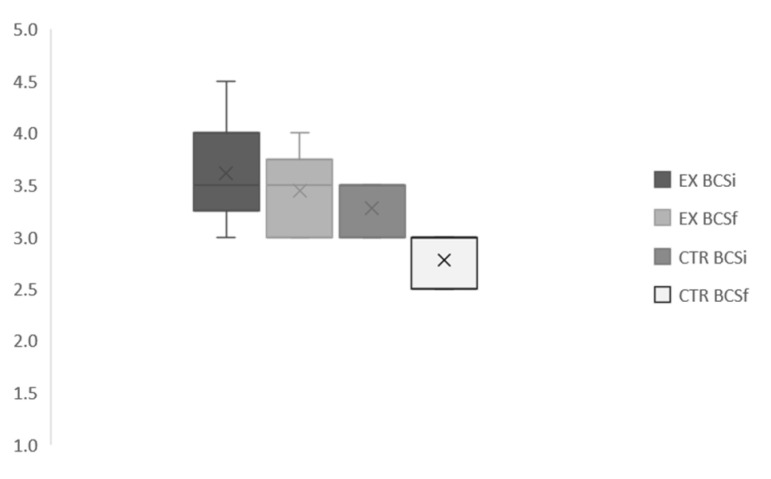
Bitches body condition score at the beginning and at the end of pregnancy. EX: experimental; CTR: control; BCS_i_: initial Body condition score; BCS_f_: body condition score after delivery.

**Figure 10 animals-11-00023-f010:**
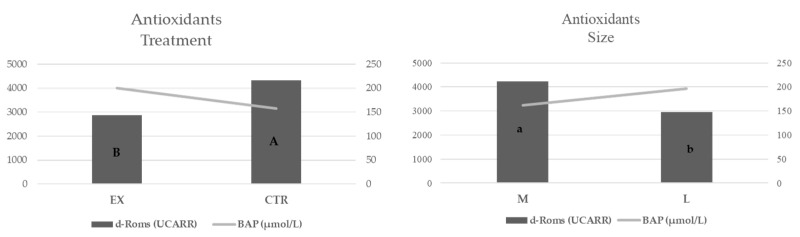
Oxidative status evaluated at the moment of pregnancy diagnosis in function of diets and size. EX: experimental group CTR: control group; d-ROMS: Reactive Oxygen Metabolites-derived compounds; BAP: Biological Antioxidant Potential. a, b: *p* < 0.05; A, B: *p* < 0.01.

**Table 1 animals-11-00023-t001:** Diets characteristics on a metabolizable energy basis (Unit/MJ ME).

Diets Characteristics	Unit	CTR	EX
Crude protein	g	17.90	19.73
Fat	g	11.26	11.28
Crude fiber	g	1.427	1.975
Ash	g	4.655	5.569
Calcium	g	0.259	0.501
Phosphorus	g	0.226	0.500
Magnesium	g	0.035	0.036
Methionine	g	0.227	0.246
Cysteine	g	0.150	0.200
Linoleic acid	g	0.717	0.806
α-Linolenic acid	g	0.024	0.049
Arachidonic acid	mg	0.016	0.018
EPA+DHA	g	0.028	0.030

CTR: Control diet (ME 16.47 MJ/kg DM); EX: Experimental diet (ME 17.04 MJ/kg DM); EPA: eicosapentaenoic acid; DHA: docosahexaenoic acid.

**Table 2 animals-11-00023-t002:** Ultrasound data.

Group	Anaestrus		Proestrus				Estrus				Ovulation	
	LA	SA	LA	SA	f/o	GR fol	LA	SA	f/o	GR fol	LA	SA
	cm	cm	cm	cm		cm	cm	cm		cm	cm	cm
Treatment												
EX	0.96 ^a^	0.64	1.77	1.06	4.27	0.46	2.17	1.15	3.84	0.79 ^a^	1.48 ^a^	0.90
CTR	0.87 ^b^	0.62	1.67	1.12	3.99	0.40	2.20	1.24	3.44	0.69 ^b^	1.21 ^b^	0.99
Size												
M	0.89	0.60	1.72	1.19 ^a^	3.82	0.42	2.14	1.19	3.55	0.73	1.33	0.90
L	0.94	0.66	1.72	0.99 ^b^	4.45	0.43	2.23	1.20	3.73	0.74	1.36	0.98
Int	**	**	NS	NS	NS	NS	NS	NS	NS	NS	NS	NS
MSE	0.007	0.009	0.028	0.027	0.618	0.021	0.102	0.019	0.196	0.010	0.040	0.002

EX: experimental diet; CTR: control diet; M: medium; L: large; Int: interaction (treatment*size); MSE: mean square error; LA: long axis; SA: short axis; f/o Follicle/ovary; GR Fol: great follicle. NS: not significantly, **: *p* < 0.01; along the column, different lowercase letters indicate difference for *p* < 0.05.

**Table 3 animals-11-00023-t003:** Monitoring of bitches until pregnancy diagnosis: vaginal cytology and serum progesterone.

Group	Proestrus		Estrus		Ovulation
	Progesterone	Sup cell	Progesterone	Sup cell	Progesterone
	Ng/mL	%	Ng/mL	%	Ng/mL
Treatment					
EX	0.83	33.52	2.89 ^a^	76.83 ^a^	6.70
CTR	0.78	35.84	2.31 ^b^	67.00 ^b^	6.78
Size					
M	0.92 ^a^	34.99	2.57	69.39 ^b^	7.08
L	0.69 ^b^	34.37	2.62	74.50 ^a^	6.40
Int	*	NS	NS	NS	NS
MSE	0.028	7.860	0.243	8.475	1.850

EX: experimental diet; CTR: control diet; M: medium; L: large; Int: interaction (treatment*size); MSE: mean square error; Sup cell: superficial cells. NS: not significantly, * *p* < 0.05; along the column, different lowercase letters indicate difference for *p* < 0.05.

## Data Availability

The data presented in this study are available on request from the corresponding author.

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
