# Peer review of "Effects of Two Commercial Diets on Several Reproductive Parameters in Bitches: Note One—From Estrous Cycle to Parturition"

_animals, 2020, doi:10.3390/ani11010023_

Round 1
Reviewer 1 Report
Thank you for your efforts in revising this manuscript. All of my concerns were addressed and I believe this is a much-improved version of an interesting topic.
Author Response
REV1
Thank you for your efforts in revising this manuscript. All of my concerns were addressed and I believe this is a much-improved version of an interesting topic.
Thank you for improving the manuscript.
Reviewer 2 Report
Overview: some typos were detected, please check again the entire paper
2.1 Animals and diets
100: please check unit for folate levels of CTR and EX diets: is it correct g/MJ ME, it seems to be much higher than the minimum recommended level for pregnancy suggested by Fediaf (12.39 microgram/MJ ME).
Table 1: why total fat content of diets is not included? It should be pertinent, as fatty acids are mentioned. Please, check the values expressed as unit/MJ ME of all nutrients of both diets, perhaps the conversion from a different unit was uncorrect; if it doesn’t, both CTR and EX are below the minimum recommended level for crude protein (12.556 CTR and 13.837 EX vs. 14.94 by Fediaf).
3 Results
Data for fetal resorption, number of puppies per litter, bitches BCS pre- and post-parturition could be expressed with graphs or tables, to better understanding, as they are presented in the text and also discussed.
4 Discussion
Comparisons with existing literature should be improved; in particular, regarding to the effects of specific nutrients or diets on numerosity of puppies for litter, as well as the effects of different dietary fatty acid concentrations during pregnancy should be discussed more in-depth, by comparing the existing literature.
Author Response
REV2
Overview: some typos were detected, please check again the entire paper
Thank you, the paper has been revised accordingly.
2.1 Animals and diets
100: please check unit for folate levels of CTR and EX diets: is it correct g/MJ ME, it seems to be much higher than the minimum recommended level for pregnancy suggested by Fediaf (12.39 microgram/MJ ME).
Sorry for the mistake, table 1 has been revised and the units were corrected.
Table 1: why total fat content of diets is not included? It should be pertinent, as fatty acids are mentioned. Please, check the values expressed as unit/MJ ME of all nutrients of both diets, perhaps the conversion from a different unit was uncorrect; if it doesn’t, both CTR and EX are below the minimum recommended level for crude protein (12.556 CTR and 13.837 EX vs. 14.94 by Fediaf).
The fat content is already reported in table 1 as “ether extract”. However, now, we had changed it in “fat”.
3 Results
Data for fetal resorption, number of puppies per litter, bitches BCS pre- and post-parturition could be expressed with graphs or tables, to better understanding, as they are presented in the text and also discussed.
The data are now reported in graphs (figures 6-9).
4 Discussion
Comparisons with existing literature should be improved; in particular, regarding to the effects of specific nutrients or diets on numerosity of puppies for litter, as well as the effects of different dietary fatty acid concentrations during pregnancy should be discussed more in-depth, by comparing the existing literature.
Thank you for your observation, the discussion has been improved citing others literature.
Reviewer 3 Report
This paper describes a study on the effects of 2 different commercial diets on several reproduction parameters in dogs. This should be reflected in the title.
The 2 diets differ in LA+ALA and in several minerals.
The discussion and conclusion of the paper mainly focus on differences in LA+ALA and in differences in EPA+DHA content (which are hardly existing) and antioxidant content (although the contents of antioxidants are not given, and no measurements of antioxidant parameters were performed). Discussion and conclusion are not taking into account differences in mineral content (which is mentioned only in the abstract).
Furthermore, not all findings were well discussed and explained in the discussion.
The paper therefore needs major revision and should focus on what has actually been done and should explain more what was the significance of the results.
Author Response
REV3
This paper describes a study on the effects of 2 different commercial diets on several reproduction parameters in dogs. This should be reflected in the title.
Thank you for your observation, the title was changed as: “Effects of two different diets on several reproductive parameters in bitches: note one - from estrous cycle to parturition”.
The 2 diets differ in LA+ALA and in several minerals.
The discussion and conclusion of the paper mainly focus on differences in LA+ALA and in differences in EPA+DHA content (which are hardly existing) and antioxidant content (although the contents of antioxidants are not given, and no measurements of antioxidant parameters were performed). Discussion and conclusion are not taking into account differences in mineral content (which is mentioned only in the abstract).
According to the suggestions of the reviewer the differences in mineral contents has been discussed. However, the antioxidant capacity was considered measuring d-Roms test (that provides a measure of the whole oxidant capacity of plasma) and BAP (the biological antioxidant potential) and were reported in figure 10. The antioxidant capacity of plasma was significantly affected by the nutritional treatment, while no differences were registered for the biological antioxidant potential.
Furthermore, not all findings were well discussed and explained in the discussion.
Thank you for your observation, the discussion has been improved citing others literature.
The paper therefore needs major revision and should focus on what has actually been done and should explain more what was the significance of the results.
The paper has been revised and improved accordingly.
Round 2
Reviewer 2 Report
All of my previous observations were taken into account and your efforts to ameliorate the manuscript are notable. However, it should be reviewed for English editing more in-depth.
Moreover, the newest part of the discussion should be improved:
- 341-343 indicate unit for protein content of the diets (% as fed/dm basis?)
- the discussion about Ca content of EX diet contrast with the M&M: since EX diet was formulated according to the aim of this study - as explained sentences 115-116 - this part of the discussion "346 Probably, for this reason, in the EX diet a higher calcium and phosphorus content was provided. 347 Indeed, such higher content was unnecessary and, also, it could interfere with the bitch ability to 348 regulate the deposition and mobilization of calcium for milk production" should be clarified.
Author Response
REV1
All of my previous observations were taken into account and your efforts to ameliorate the manuscript are notable. However, it should be reviewed for English editing more in-depth.Moreover, the newest part of the discussion should be improved:
- 341-343 indicate unit for protein content of the diets (% as fed/dm basis?)
Thank you for your observation, the percentage of protein content as expressed in % as fed and the indication was added in the text.
- the discussion about Ca content of EX diet contrast with the M&M: since EX diet was formulated according to the aim of this study - as explained sentences 115-116 - this part of the discussion "346 Probably, for this reason, in the EX diet a higher calcium and phosphorus content was provided. 347 Indeed, such higher content was unnecessary and, also, it could interfere with the bitch ability to 348 regulate the deposition and mobilization of calcium for milk production" should be clarified.
According to your suggestion, we changed the sentence in order to better clarify the concept.
Reviewer 3 Report
Thank you for addressing my concerns. I am still lacking more in depth explanations on differences in mineral content, and fatty acid patterns and how this would improve reproductive outcome. E.g. Essential fatty acids and trace minerals have various effects on ovarian hormone production, uterine protein production, placentation and fetal development. Is a rather general statement that demands further explanation. E.g. Another nutritional factor which may have favored the faster embryo growth could be the different Linoleic and α-Linolenic ratios (LA:ALA). Needs further explanation. The EX diet showed higher concentrations of Linoleic, α-Linolenic acids and Eicosapentaenoic acid (EPA) plus Docosahexaenoic acid (DHA). I strongly disagree that the amounts of EPA+DHA were higher in such a way that it would affect reproductive outcome.
Author Response
REV2
Thank you for addressing my concerns. I am still lacking more in depth explanations on differences in mineral content, and fatty acid patterns and how this would improve reproductive outcome. E.g. Essential fatty acids and trace minerals have various effects on ovarian hormone production, uterine protein production, placentation and fetal development. Is a rather general statement that demands further explanation. E.g. Another nutritional factor which may have favored the faster embryo growth could be the different Linoleic and α-Linolenic ratios (LA:ALA). Needs further explanation. The EX diet showed higher concentrations of Linoleic, α-Linolenic acids and Eicosapentaenoic acid (EPA) plus Docosahexaenoic acid (DHA). I strongly disagree that the amounts of EPA+DHA were higher in such a way that it would affect reproductive outcome.
Considering the relative low differences between the diets in mineral content we focalized the discussion on Ca and P levels. Indeed, the sentence was changed according to the suggestion of Reviewer 1 (please, see lines 260-367), and now we think it is more understandable.
Regarding essential fatty acids we think to have clarified the role of these fatty acids in prostaglandins and hormone synthesis (lines 320-331).
Moreover, concerning your comment on the role of EPA and DHA, we are agree with you, we didn’t think that the dietary amounts of these fatty acids have contributed in reproductive outcome, but that these components of the diets could contribute to exert an antioxidant activity, as evidenced by the significant lower values of d-ROMs test (that measure the whole oxidant capacity of plasma) founded in EX diet compared to CTR one. We have clarified this concept in the discussion and the possible role of EPA and DHA in the diets.
This manuscript is a resubmission of an earlier submission. The following is a list of the peer review reports and author responses from that submission.
Round 1
Reviewer 1 Report
This paper cannot be evaluated as a scientific paper without knowing anything about the two foods - there is not enough information regarding ingredients and all nutrient levels.
The measurements used in this study have a fatal flaw and no conclusions can be made - the investigators make an assumption that all breeds are the same. They are not regarding follicular size and number. Similarly the measuring of the fetus cannot be done across breeds. This is a fatal flaw and this paper should not be accepted due to these fatal flaws
Reviewer 2 Report
This is an interesting and comprehensive study on the effect of a high protein diet on certain aspects of estrous and fetal development in dogs. Strengths of this study include the close monitoring with several different methods (ultrasound, blood tests, vaginal cytology, markers of oxidative stress, and body weight/condition scores. Limitations of the manuscript include a high proportion of citations being from review articles vs. original research sources, and missing some statistical analysis, specifically regarding difference in fetal absorption percentages. There are several minor issues with English grammar which make some statements unclear.
Line 159: "clore" likely refers to chloride.
Line 186 "Considering the bitches size during the anestrus, ovarian were significantly..." something is missing immediately after "ovarian," and "bitches" should be bitches' with a terminal apostrophe if this is meant to be plural possesive.
Lines 188-190: "In any case the highly significant interaction (treatment*size, <0.001) observed for both ovarian axes in anestrus testify a not linear trend into the groups." is unclear. It seems to be referencing the statistical significance of the observations for ovarian axes size in anestrus, stating it demonstrated a non-linear trend, but "into the groups" is not clear. Assistance with English translation should clarify this apparently important statement.
Regarding citations/references: In the entire introduction, where the authors cite 9 individual references, only two of these references (6,7) are original research, with the rest being reviews. If the authors are citing evidence-based information they obtained from the review articles, they should cite the original research instead of the review that included that research. If they are citing the conclusions or opinions of the review authors, they should indicate that this was the review authors' opinion vs. original research.
Lines 88-93: Regarding Materials and Methods, describing the make-up of the different large and medium sized dogs in the control and experimental groups: There seems to be an imbalance in the numbers of medium and large dogs in the CTR and EXP group. There were 5 large dogs in the CTR group and 7 in the EXP group. Why were they not randomized into 6 and 6 dogs in each of the CTR and EXP groups? Similarly, there are only two medium sized dogs in the EXP group and 4 in the CTR group. Why not 3 and 3, respectively? Were there additional dogs that were eventually excluded or lost to follow-up? I would appreciate the authors providing and explanation for how they constructed the groups to end up with these uneven numbers, when it seems possible (based on the numbers, alone) that even numbers could have been assigned to each group.
Line 97: CP, EE, Ca, should be spelled out in full the first time they are used in the manuscript, followed by the abbreviation, or provide a table with abbreviation definitions somewhere in the manuscript.
Lines 104-105: The authors describe that in the last third of gestation the dogs were allowed access to the food all day. Was the amount that they consumed measured or quantified in any way? This could have an important impact on the amount of each nutrient they received on the different diets so knowing how much each dog ingested and factoring that into nutritional quantities would be important to fully understand the impact of each diet on the factors studied.
Lines 200-202/Figure 2: It seems reasonable that dogs with larger body size would have larger inner chorionic size (diameter). Is there any way to normalize this data to account for dog body size? For example, analyzing the ratio of inner chorionic diameter to body weight, or something similar?
Lines 226-229: Reabsorption percentage was less for EXP group compared to CTR, but I don't see where the authors provide any statistical analysis on whether this difference was significant. Statistical significance should be demonstrated for us to accept that there was any true difference between the EXP and CTR groups.
Line 256: "trough" looks like it should be "through."
Lines 263 and 265: It appears that citation 20 comes before citation 19 appears in the manuscript. Also, citation is a review article. Authors should cite the original source of research vs. the review article.
Lines 231-293: Citation 32 is a review article. Please cite the original research instead.
Lines 295-298: "In conclusion our study evidenced the importance of a continuous monitoring of pregnancy in dog in order to assess bitches and fetal welfare." This was not the purpose of your study, nor did you compare outcomes of bitch or fetal welfare as a result of continuous monitoring vs. no continuous monitoring. So while you did use continuous monitoring, it was to assess the potential differences in bitch and fetal welfare in dogs on two separate diets. The opening statement of your conclusion should directly address the main findings of the variables you studied (CTR and EXP diet). Your study did not propose to evaluate the benefit of continuous monitoring, nor did it demonstrate its benefit, (compared to not doing it), so you cannot make any conclusions on its importance. I recommend that you reword your conclusions section to emphasize the primary findings of your study, and then if you wish, follow that with commentary on how the continuous monitoring helped your data collection and may be a beneficial tool in evaluating pregnancy progress in dogs.
Reviewer 3 Report
Text and English language editing should be completely revised, as many grammar and formal errors occured. Also, please pay attention to standardise terminology (bitches or dams? Delivery or parturition? …and so on).
Introduction: please note that Ca, P and Mg are not trace elements.
Materials and methods:
- Prefer body weight for companion animals instead of live weight
- I have a major concern about dog breeds division into groups: in the CTR group English Bulldog - please note that is not “bulldogue”- is considered a medium size dog, while in the EX group, the 2 English bulldog were split, 1 into medium size group the other one into large size group, and I don’t think it make sense.
Moreover animals involved were not grouped uniformly, so that the EX group had a much larger number of large size dogs, compared to medium size dogs, and it cannot be ruled out that difference could might has affected the results.
- When describing the composition of experimental diets prefer g/Mcal or MJ
- Please check and correct the Ca content of EX diet
- remove the name of referral clinic, it’s not pertinent
- Tables 1 and 2 are not pertinent – and also the Table 1 legend is uncorrect- and do not provide any contributions to the aim of the research (please check and correct the text abbreviations for mineral blood profile)
The main issue is that 2 experimental diets, as presented, differ little from each other and in the discussion part is not explained and well developed the reasons why the EX diet should be beneficial to EX group as opposed to CTR group.
Moreover, the discussion part should be reviewed in-depth, as is lacking of comparisons with existing literature.